# Impact of New Bed Assignment Information System on Emergency Department Length of Stay: An Effect Evaluation for Lean Intervention by Using Interrupted Time Series and Propensity Score Matching Analysis

**DOI:** 10.3390/ijerph19095364

**Published:** 2022-04-28

**Authors:** Chih-Chien Yun, Sin-Jhih Huang, Tsuang Kuo, Ying-Chun Li, Wang-Chuan Juang

**Affiliations:** 1Department of Emergency, Kaohsiung Veterans General Hospital, Kaohsiung 813414, Taiwan; ccyun@vghks.gov.tw; 2Quality Management Center, Kaohsiung Veterans General Hospital, Kaohsiung 813414, Taiwan; cingchih@gmail.com; 3Department of Business Management, National Sun Yat-sen University, Kaohsiung 804201, Taiwan; kuo@bm.nsysu.edu.tw; 4Institute of Health Care Management, National Sun Yat-sen University, Kaohsiung 804201, Taiwan; ycli@mail.nsysu.edu.tw

**Keywords:** emergency department crowding, length of stay, lean, interrupted time series, intervention

## Abstract

A long waiting period for available beds in emergency departments (EDs) is the major obstacle to a smooth process flow in ED services. We developed a new bed assignment information system that incorporates current strategies and resources to ease the bottleneck in the service flow. The study’s purpose was to evaluate the effect of the lean intervention plan. We included 54,541 ED patient visits in the preintervention phase and 52,874 ED patient visits in the postintervention phase. Segmented regression analysis (SRA) was used to estimate the level and trend in the preintervention and postintervention phases and changes in the level and trend after the intervention. After the intervention, the weekly length of stay (LOS) for patient visits, admitted patient visits, and nonadmitted patient visits decreased significantly by 0.75, 2.82, and 0.17 h, respectively. The trendline direction for overall patient visits and nonadmitted patient visits significantly changed after the intervention. However, no significant change was noted for admitted patient visits, although the postintervention trend visually differed from the preintervention trend. The concept of lean intervention can be applied to solve various problems encountered in the medical field, and the most common approach, SRA, can be used to evaluate the effect of intervention plans.

## 1. Introduction

The emergency department (ED) is a vital component of the healthcare system and provides emergency care 24/7 [1]. However, overcrowding in EDs has become a crucial public health problem worldwide in the past decade, including in Taiwan [2]. The high workload results in failure or delay of service delivery, increased length of stay (LOS) of patients, delayed clinical decision-making, and even irreversible medical errors as a result of wrong decisions [3,4]. These factors further reduce the quality of care, staff work efficiency, and patient satisfaction [3,5].

Systematic reviews have demonstrated that factors contributing to ED-related crowding [3,6] can be divided into three categories: input, throughput, and output [7]. In general, crowding is affected by the number of patients, medical staff, and available beds in the ED as well as by wait times for laboratory tests and radiology examinations [8]. However, increasing evidence indicates that the boarding of admitted patients might be the primary cause of ED overcrowding [9,10,11]. Therefore, our study focused on the output category of influencing factors, especially among admitted patients waiting for available beds. During their stay, admitted patients are often kept in the ED if no suitable or designated space is available in other wards. Various factors contribute to inpatient boarding in the ED, including the lack of inpatient beds, inefficient diagnostic and ancillary services in inpatient units, late discharge on the scheduled discharge day, and delays in cleaning rooms after patient discharge [7]. Inpatient boarding in the ED does not necessarily indicate crowding in the ED but indicates general hospital crowding. This crowding implies a demand-and-supply imbalance of health services.

In 2016, we first implemented quality control circle activities in an attempt to solve the ED overcrowding problem [12]. Our survey results showed that up to 90% of patients visiting the ED stayed for more than 48 h because of the lack of available beds. We then increased the turnover rate of inpatient beds to decrease the ED LOS by emphasizing the discharge before noon policy for all inpatients. In addition, drawing from the Plan-Do-Check-Act cycle, an iterative four-step management method used in various industries for the control and continuous improvement of processes and products (i.e., there is no best, only better), we conducted this study to increase the efficiency of patient bed management. Earlier, bed assignment and management were conducted over the phone; this strategy was prone to delays in the process caused by delayed bed cleaning and clearance. Therefore, we commissioned the hospital information department to design a new bed assignment information system (BAIS) that incorporated current strategies and links other existing information systems.

Flow is a key concept of lean that is derived from the Toyota Production System. A major obstacle to creating a smooth flow is the bottlenecks encountered in the service process, which were the long waiting times for available beds in this context. Therefore, this study evaluated the effect of the lean intervention plan by using an interrupted time series (ITS) design. This approach was used to strengthen the before-and-after design and enabled the statistical investigation of potential biases in the estimation of the intervention effect [13].

## 2. Materials and Methods

### 2.1. Data Collection

ED patients’ visit data were retrospectively collected from the hospital information system of a medical center in Southern Taiwan by using the WebFOCUS tool (Information Builders, New York, NY, USA). The center had a total of 53 beds in the ED at the beginning of the project, including 5 beds in the pediatric emergency area, 10 beds in the resuscitation area, 10 beds in the trauma area, 3 beds in the isolation area, and the remaining beds in the observation area. Notably, the ED also provides an area for patients with less severe medical conditions, such as acute allergic reactions or an acute gout attack, to sit upright.

Considering that the effect of the intervention introduced in 2017 may take time to present, we divided the patients visiting the ED in 2016 and 2018 into two groups, the preintervention group (PREG) and the postintervention group (POSG), respectively. To maintain the high homogeneity of analytical data, patients who had undergone trauma, were younger than 20 years, were visiting for children’s emergency care, or had missing or incomplete information were excluded. A total of 54,541 PREG and 52,874 POSG patient visits that met the established criteria were included in the analysis. After the composition of the study population and medical information was balanced through propensity score matching between the PREG and POSG, 89,818, 20,974, and 68,844 patient visits were included in the analysis to evaluate the lean intervention effect for ED patient visits, ED admitted patient visits, and ED nonadmitted patient visits, respectively.

### 2.2. Bed Assignment Information System

Bed assignment requires special training and the simultaneous consideration of many parameters and situations. When bed assignment was conducted manually in the past, the staff contacted the wards bidirectionally by telephone to obtain all the bed numbers that were scheduled for discharge. Beds must be assigned on the basis of the information collected and in accordance with the established policy of sequentially assigning patients to available beds. To reduce the disruption of telephone hotlines and the burden on administrative staff as well as improve the efficiency of bed arrangements, the information department developed a new BAIS in 2017 and implemented it by the end of the same year. Through Web technology, the BAIS is deeply connected with the nursing, ED, and hospital information systems. The BAIS solved the problem of simultaneous bed information aggregation on the supply and demand sides. Moreover, this system features a complementary function of bed-needs-cleaning notifications, through which the cleaning staff is informed of which beds require cleaning on demand. Once the cleaning task is completed, feedback is sent online to the BAIS so that the BAIS operator can assign the available cleaned bed to an ED admitted patient in need.

### 2.3. Statistical Analysis

Descriptive statistics for continuous and categorical variables are reported as means ± standard deviation (SD) and frequencies (percentages), respectively. Continuous variables were compared using Student’s *t*-test, and categorical variables were compared using the Chi-square test. A 1:1 propensity score matching method was used to balance the composition of the study population and medical information between the PREG and POSG using PROC PSMATCH, a procedure of the SAS/STAT module. The matched variables between the PREG and POSG included age, patient seniority (older or younger patients), sex, month of visit, triage, disease, and patient disposition. The elderly population included people aged 65 and over.

Segmented regression, which typically aggregates individual-level data by time point, was applied in this study. To maintain a sufficient number of preintervention and postintervention data points and to obtain adequate power to estimate regression coefficients, the continuous outcome in the time series was replaced by the weekly LOS, which was determined by the median LOS at every ED visit for all 7 days of the week instead of the monthly LOS. The LOS was calculated as the time between registration and discharge. Aggregation was calculated using the PROC TIMESERIES. The detection and lag length of the autocorrelation in successive time-series observations were assessed using the Durbin–Watson test in PROC AUTOREG and PROC ARIMA, respectively. The cyclical signal in the time series was evaluated using PROC ARIMA.

Subsequently, in this ITS with 53 preintervention data points and 52 postintervention data points, segmented linear models were used to estimate the level and trend in the preintervention and postintervention segment and the changes in the level and trend after the intervention. Segmented regression analysis (SRA) was conducted using an SAS macro developed by Dr. Joseph M. Caswell [14]. However, once an autocorrelation occurred, we adjusted the standard errors using the Newey–West estimator in PROC MODEL during SRA to avoid the underestimation of standard errors and the recording of inaccurate significant effects. The aforementioned four procedures are a part of the SAS/ETS module. The best-fit preintervention and postintervention regression lines were estimated and generated by PROC SGPLOT, a procedure in SAS/GRAPH. All statistical analyses and plotting were performed using SAS v9.4m7 (SAS Institute Inc., Cary, NC, USA). Differences with a two-tailed *p*-value of <0.05 were considered statistically significant.

## 3. Results

### 3.1. Comparison of the Composition of the Study Population and Medical Information between PREG and POSG

A total of 89,818 ED patient visits (comprising 44,909 visits in the PREG and 44,909 visits in the POSG), 20,974 ED admitted patient visits (comprising 10,487 visits in the PREG and 10,487 visits in the POSG), and 68,844 ED nonadmitted patient visits (comprising 34,422 visits in the PREG and 34,422 visits in the POSG) were included for evaluating the effect of the lean intervention. The visits in the PREG were matched with those in the POSG in terms of the aforementioned variables, but the patients in the POSG had a slightly higher age than those in the PREG (*p* = 0.159, *p* = 0.340, and *p* = 0.422 for all patient visits, admitted patient visits, and nonadmitted patient visits, respectively). The distribution and proportion of other variables were the same in the PREG and POSG (*p* = 1.000 for all patient visits, admitted patient visits, and nonadmitted patient visits; Table 1).

### 3.2. Comparison of Weekly LOS for ED Patient Visits between the Preintervention and Postintervention Phases

Our results showed that the cyclical signal in the time series was not found. This implied that preprocessing was not required before entering the main analysis. However, the Durbin–Watson test revealed autocorrelation in the time series (Durbin–Watson D statistic = 1.20, *p* < 0.001 for a positive autocorrelation). The partial autocorrelation function (PACF) plot, drawn by PROC ARIMA, is useful in identifying the order of an autoregressive model. The order is the minimum number of lags with a spike beyond the twofold standard error (that is, the significant spike). The PACF plot indicated that the order was 1. The outcome was consistent with the autocorrelation function (ACF) plot that was generated for every nonsignificant spike, except for the spike at lag = 0 after controlling for the number of lags (lag = 1). Because of the presence of the autocorrelation, SRA with the Newey–West estimator was performed (with order as an important parameter). The results of SRA with the Newey–West estimator revealed a significant preintervention trend, with the weekly LOS increasing by approximately 0.016 h per week (*p* < 0.001). After the preintervention trend was accounted for, a significant postintervention level change was observed, with the weekly LOS decreasing by approximately 0.75 h (*p* < 0.001). The postintervention trend significantly differed from the preintervention trend (*p* = 0.009), although the postintervention trend was not significant (*p* = 0.712). Figure 1 illustrates the level and trend of the weekly LOS for ED patient visits with fitted regression lines during the preintervention and postintervention phases.

### 3.3. Comparison of Weekly LOS for ED Admitted Patient Visits between the Preintervention and Postintervention Phases

Similarly, a cyclical signal in the time series was not found. However, the Durbin–Watson test revealed an autocorrelation in the time series (Durbin–Watson D statistic = 1.35, *p* < 0.001 for a positive autocorrelation). The PACF plot indicated that the order was 1. This outcome was consistent with that of the ACF plot that was generated for every nonsignificant spike, except for the spike at lag = 0 after controlling for the number of lags (lag = 1). Because of the presence of autocorrelation, SRA with the Newey–West estimator was conducted. The results of SRA with the Newey–West estimator revealed a nonsignificant preintervention trend for the weekly LOS (*p* = 0.649). After the preintervention trend was accounted for, a significant postintervention level change was observed, with the weekly LOS decreasing by approximately 2.82 h (*p* = 0.023). The postintervention trend differed from the preintervention trend, but the difference was not statistically significant (*p* = 0.117). Finally, we noted a significant postintervention trend, with the weekly LOS decreasing by approximately 0.06 h per week (*p* = 0.032). Figure 2 depicts the level and trend of the weekly LOS for ED admitted patient visits with fitted regression lines during the preintervention and postintervention phases. Moreover, the overlay histograms clearly show that the distribution of the weekly LOS shifted to the lower LOS zone after the implementation of the intervention (Figure 3).

### 3.4. Comparison of Weekly LOS for ED Nonadmitted Patient Visits between the Preintervention and Postintervention Phases

Similarly, a cyclical signal in the time series was not found. However, the Durbin–Watson test revealed an autocorrelation in the time series (Durbin–Watson D statistic = 1.55, *p* = 0.008 for a positive autocorrelation). The PACF plot indicated that the order was 1. This outcome was consistent with the ACF plot that was generated for every nonsignificant spike, except for the spike at lag = 0 after controlling for the number of lags (lag = 1). Because of the presence of autocorrelation, SRA with the Newey–West estimator was conducted. The results of SRA with the Newey–West estimator revealed a significant preintervention trend, with the weekly LOS increasing by approximately 0.005 h per week (*p* = 0.014). After the preintervention trend was accounted for, a significant postintervention level change was noted, with the weekly LOS decreasing by approximately 0.17 h (*p* = 0.013). The postintervention trend differed from the preintervention trend, but not significantly (*p* < 0.001). A significant postintervention trend was also noted, with the weekly LOS decreasing by approximately 0.004 h per week (*p* = 0.022). Figure 4 illustrates the level and trend of the weekly LOS for ED nonadmitted patient visits with fitted regression lines during the preintervention and postintervention phases.

## 4. Discussion

### 4.1. Statement of Principal Findings

This study evaluated the effect of a lean intervention by using an ITS design. Through SRA, the most used method in ITS, we confirmed that the lean intervention plan of developing a BAIS that incorporates current strategies and links other existing information systems was effective. Our results indicated an apparent immediate decrease (level change for the weekly LOS) following the intervention implementation as well as a decrease in the trend (slope change for the weekly LOS). In addition, the intervention reduced the congestion pressure on the ED workflow that was caused by the previous bottleneck. This improvement plan was aimed at ED-admitted patients. However, after the introduction of the intervention in 2017, we found that the weekly LOS of ED-nonadmitted patients also decreased significantly. Finally, we found that the weekly LOS of all ED patients significantly decreased following the intervention. After the trend was controlled for, a significant level change following the intervention was noted, with the weekly LOS decreasing by approximately 0.75, 2.82, and 0.17 h for ED patient visits, ED admitted visits, and ED nonadmitted visits, respectively. Thus, this intervention plan reduced the maximum LOS for ED admitted visits. In addition, following the intervention, a significant downward trend was observed for ED admitted visits and ED nonadmitted visits.

### 4.2. Strengths and Limitations

This study has several notable strengths. First, more than 65% of previous studies have performed analyses inappropriately. Ramsay et al. suggested that most of these studies have performed *t*-tests of the preintervention points versus postintervention points. The *t*-test yields incorrect effect sizes if a preintervention trend is present and decreases the standard error if a positive autocorrelation is present [13]. By contrast, we used the ITS design to estimate the effect of the lean intervention. For the preintervention and postintervention phase segments of a time series, the level and trend values are determined using the SRA approach. Compared with traditional before-and-after studies, ITS designs are generally more robust and are arguably considered the optimal approach for evaluating the effect of hospital-wide interventions and new policies implemented nationwide [15]. Second, Penfold et al. suggested that a minimum of eight time points per period is required to achieve sufficient power when estimating regression coefficients [16]. In our study, the sequence length in the preintervention and postintervention periods comprised up to 53 and 52 data points, respectively. In addition, a long preintervention phase was beneficial, thereby increasing the power to detect secular trends [13]. Third, because time is a predictor in SRA, error terms of consecutive observations are often correlated. The correlation between data points is termed autocorrelation. Failure to correct for the autocorrelation may lead to the underestimation of standard errors and overestimation of the effect significance of an intervention [17]; therefore, we corrected for the autocorrelation in this study. Fourth, Wagner et al. considered that when using monthly time-series data, at least 12 data points before and after the intervention are recommended to meaningfully adjust for seasonality [17]. On the basis of this concept, our ITS design with 53 weekly LOS preintervention data points in 2016 and 52 weekly LOS postintervention data points in 2018 can be regarded as deseasonalized time-series data. In addition, no cyclical signal was found in the time series. The aforementioned confounders were potential biases [13]. Over 40% of studies have not tested or accounted for potential biases in the data because of, for instance, autocorrelation, seasonality, or heteroskedasticity effects [15]. Fifth, the composition of the study population and medical information may change the weekly LOS during the preintervention and postintervention phases. Therefore, before aggregating individual-level data, we matched seven variables between the PREG and POSG. Finally, we confirmed that the change in the weekly LOS during the postintervention phase was not caused by these confounders. Studies that perform matching between the preintervention and postintervention periods prior to ITS analysis are rare. Sixth, Ramsay et al. described the quality criteria for ITS designs, and our design met seven of eight requirements [13]. The criterion that was not met was item 7: “the shape of the intervention effect was prespecified.” This means a rational explanation for the shape of the intervention effect was given by the author(s). Because ITS designs should not include the interpretation of statistical results, the item was an adequate quality criterion for reviewing related ITS design articles. We assessed whether 58 ITS studies met the criteria and found that none of the studies met the requirements.

A limitation of this study is that our ITS design lacked comparative control time-series data. The purpose of including control data in the design is to exclude time-varying confounders, especially for cointerventions or other events that occur during the intervention, as these generally cannot be predicted on the basis of modeling preintervention trends [18]. However, assessing the impact of policy changes, especially improvement plans in hospitals, is rarely possible with such study designs [17,19]. For example, to reduce the proportion of ED patients who stayed for more than 48 h, the hospital team implemented a series of improvement measures. Selectively implementing a measure for certain types of ED patients with the aim of collecting comparative control time-series data is not possible.

### 4.3. Strengths and Weaknesses in Relation to Other Studies, Discussing the Important Differences in Results

We described the strengths and weaknesses of our study. Regarding the composition of the study population, because the boarding of admitted patients might be the primary cause of ED overcrowding (as stated in the Introduction section), the distributions of patient disposition for ED patient visits were matched with the same result between the preintervention and postintervention periods (see Table 1). The intervention effect was not likely to be related to the different distribution of patient dispositions. The overall mean weekly LOS of ED patient visits slightly decreased from 3.75 h before the intervention to 3.45 h after the intervention. Lee et al. demonstrated that the mean ED LOS for ED patient visits decreased from 9.47 h in the preintervention period to 5.76 h after the intervention [20]. This decrease was larger than that in our study. However, their table showed that the frequency of admission was 26.9% in the preintervention and 21.9% in the postintervention, which may result in an excessive decrease. In addition, although the intervention plan led to a decrease in level, an upward trend unexpectedly occurred in the postintervention period (monthly LOS increased by approximately 0.18 h per month (*p* < 0.001)). This indicates that LOS may not be sustained over time without further intervention, and the effect may be caused by other factors. This phenomenon was not found in our study for ED patient visits, ED admitted patient visits, or ED nonadmitted patient visits.

### 4.4. Meaning of the Study: Possible Explanations and Implications for Clinicians and Policymakers

One of the analysis rules often adopted by quality improvement teams is Pareto’s principle, particularly in the context of quality-related issues. Pareto’s principle is also known as the 80/20 rule and simply means that in many cases, 80% of the effects come from 20% of the causes [21]. For any improvement, most of the efforts must be focused on the most contributing factors. As mentioned above, up to 90% of the patients visiting EDs stayed for more than 48 h because of the lack of available beds. These results were consistent with those of Alemu et al. [22], who demonstrated that the inadequacy of beds in inpatient wards induced a significant difference in an LOS greater than 24 h compared with an LOS ≤ 24 h in EDs (adjusted odds ratios: 8.7, the highest value in the list of factors related to LOS). Brainstorming can easily guide a team to focus on improvement measures, including avoiding delays in bed cleaning [7]; emphasizing early inpatient discharge on the same day [12,23]; strengthening communication regarding discharge and bed availability [24]; providing education for all clinical staff regarding new bed management policies [24]; introducing a bed manager who can assess bed availability in real-time and triage and admit patients to inpatient beds [25]; and introducing a bed director who can utilize other resources and measures, including extra staff or admission of medical patients to nonmedical beds, to avoid a temporary overwhelming overload [25]. The common purpose of these strategies is to increase the turnover rate for inpatient beds so that a patient in the ED waiting for an available bed can be immediately admitted to the ward. In this study, we proved through the SRA method that integrating existing operational strategies, connecting existing systems, and connecting upstream and downstream processes more compactly via the BAIS can also achieve significant improvements.

Hospitals must be regularly evaluated and certified by health authorities. Most of the accreditation provisions require that the intervention plans for improvement should include a science-based effect evaluation. This can be achieved using information technology; however, objective data analysis and corroboration are necessary. Our study employed the SRA method in ITS to evaluate the longitudinal effects of the intervention plan and have attempted to identify the strategies and resources to significantly alleviate the existing bottleneck in patient flow.

### 4.5. Unanswered Questions and Future Research

Multiple academicians and practitioners have attempted to define the concept of lean [26]. Toussaint and Berry defined it as a cultural transformation that changes how an organization works [27]. Burgess and Radnor stated that lean involves improving quality to eliminate activities that add no value [28]. Abdi et al. considered it to be improved utilization of the organization’s resources [29]. Modig and Ahlstrom described it as an operations strategy that prioritizes flow efficiency over resource efficiency [30]. In summary, the concept of lean involves the transformation of key processes with minimal resources (i.e., manpower and time) to achieve a smooth service flow and ease congestion pressure. We used this concept to solve the current problems encountered in the medical field and applied SRA, the most common approach, to evaluate the effect of the intervention plan. The practical operation of the BAIS is beyond the scope of discussion in this study.

## 5. Conclusions

The BAIS designed in this study, which incorporates current strategies, links other existing information systems, and connects upstream and downstream processes more compactly, may exert a synergistic effect on the alleviation of the obstacles to patient flow. The lean concept can be adopted to solve various existing problems in the medical field. We used the most common approach, SRA, to evaluate the effect of our intervention plan. However, considering the presence of a control group with a change similar to that in the intervention group, the decrease may have been due to some other event or co-intervention that affected both groups.

## Figures and Tables

**Figure 1 ijerph-19-05364-f001:**
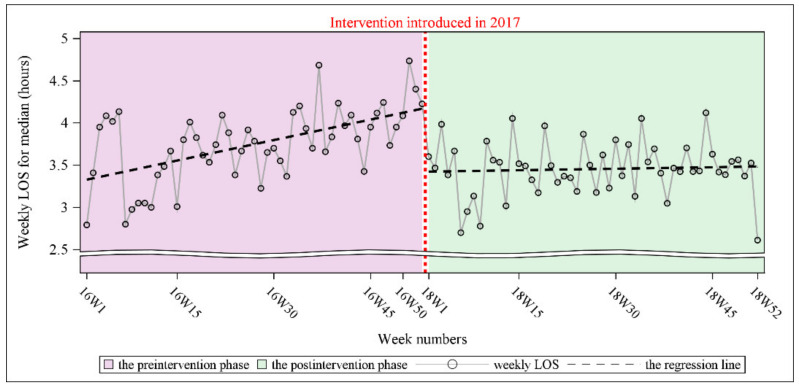
Comparison of weekly LOS for ED patient visits between the preintervention and postintervention phases with fitted segmented regression lines.

**Figure 2 ijerph-19-05364-f002:**
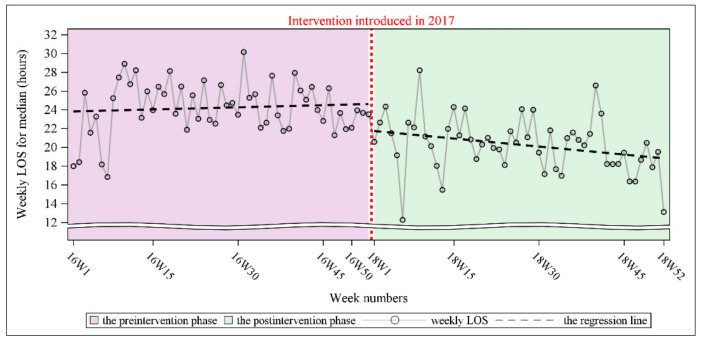
Comparison of weekly LOS for ED admitted patient visits between the preintervention and postintervention phases with fitted segmented regression lines.

**Figure 3 ijerph-19-05364-f003:**
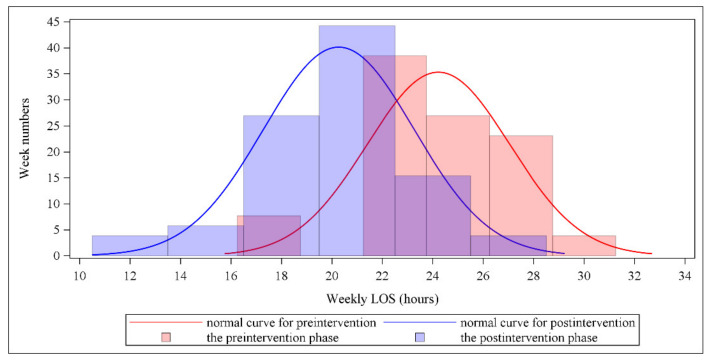
Comparison of weekly LOS for ED admitted patient visits between the preintervention and postintervention phases.

**Figure 4 ijerph-19-05364-f004:**
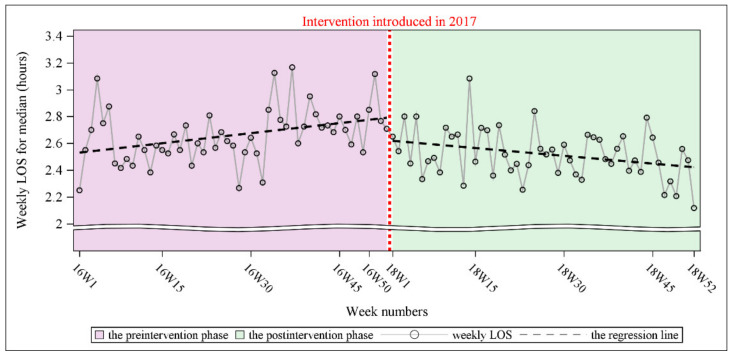
The comparison of weekly LOS for ED nonadmitted patient visits between the preintervention phase and the postintervention phase.

**Table 1 ijerph-19-05364-t001:** Comparison of demographic and medical information for ED patient visits between the PREG and POSG.

Factors/Category	All Patient Visits (N = 89,818)		Admitted Patient Visits (N = 20,974)		Non-Admitted Patient Visits (N = 68,844)	
	PREG N = 44,909 n (%)	POSG N = 44,909 n (%)	PREG N = 10,487 n (%)	POSG N = 10,487 n (%)	PREG N = 34,422 n (%)	POSG N = 34,422 n (%)
Age mean ± SD (years)	56.5 ± 20.1	56.7 ± 20.0	66.5 ± 17.9	66.7 ± 17.7	53.5 ± 19.7	53.6 ± 19.6
Nonelder	28,175 (62.7)	28,175 (62.7)	4457 (42.5)	4457 (42.5)	23,718 (68.9)	23,718 (68.9)
Elder	16,734 (37.3)	16,734 (37.3)	6030 (57.5)	6030 (57.5)	10,704 (31.1)	10,704 (31.1)
Sex						
Female	21,179 (74.2)	21,179 (74.2)	4121 (39.3)	4121 (39.3)	17,058 (49.6)	17,058 (49.6)
Male	23,730 (52.8)	23,730 (52.8)	6366 (60.7)	6366 (60.7)	17,364 (50.4)	17,364 (50.4)
Month						
1	3712 (8.3)	3712 (8.3)	870 (8.3)	870 (8.3)	2842 (8.3)	2842 (8.3)
2	4545 (10.1)	4545 (10.1)	848 (8.1)	848 (8.1)	3697 (10.7)	3697 (10.7)
3	4042 (9.0)	4042 (9.0)	830 (7.9)	830 (7.9)	3212 (9.3)	3212 (9.3)
4	3713 (8.3)	3713 (8.3)	855 (8.2)	855 (8.2)	2858 (8.3)	2858 (8.3)
5	3677 (8.2)	3677 (8.2)	893 (8.5)	893 (8.5)	2784 (8.1)	2784 (8.1)
6	3636 (8.1)	3636 (8.1)	869 (8.3)	869 (8.3)	2767 (8.0)	2767 (8.0)
7	3705 (8.3)	3705 (8.3)	880 (8.4)	880 (8.4)	2825 (8.2)	2825 (8.2)
8	3642 (8.1)	3642 (8.1)	883 (8.4)	883 (8.4)	2759 (8.0)	2759 (8.0)
9	3568 (7.9)	3568 (7.9)	868 (8.3)	868 (8.3)	2700 (7.8)	2700 (7.8)
10	3662 (8.2)	3662 (8.2)	869 (8.3)	869 (8.3)	2793 (8.1)	2793 (8.1)
11	3391 (7.6)	3391 (7.6)	883 (8.4)	883 (8.4)	2508 (7.3)	2508 (7.3)
12	3616 (8.1)	3616 (8.1)	939 (9.0)	939 (9.0)	2677 (7.8)	2677 (7.8)
Triage						
1	1530 (3.4)	1530 (3.4)	1190 (11.4)	1190 (11.4)	340 (1.0)	340 (1.0)
2 or 3	42,609 (94.8)	42,609 (94.8)	9279 (88.5)	9279 (88.5)	33,330 (96.8)	33,330 (96.8)
4 or 5	770 (1.7)	770 (1.7)	18 (0.2)	18 (0.2)	752 (2.2)	752 (2.2)
Disease						
Complications of pregnancy, childbirth, and the puerperium	2 (<0.1)	2 (<0.1)	1 (<0.1)	1 (<0.1)	1 (<0.1)	1 (<0.1)
Congenital anomalies	18 (<0.1)	18 (<0.1)	3 (<0.1)	3 (<0.1)	15 (<0.1)	15 (<0.1)
Diseases of the blood and blood-forming organs	116 (0.3)	116 (0.3)	29 (0.3)	29 (0.3)	87 (0.1)	87 (0.1)
Diseases of the circulatory system	3502 (7.8)	3502 (7.8)	1304 (12.4)	1304 (12.4)	2198 (6.4)	2198 (6.4)
Diseases of the digestive system	4851 (10.8)	4851 (10.8)	1195 (11.4)	1195 (11.4)	3656 (10.6)	3656 (10.6)
Diseases of the genitourinary system	3761 (8.4)	3761 (8.4)	1143 (10.9)	1143 (10.9)	2618 (7.6)	2618 (7.6)
Diseases of the musculoskeletal system and connective tissue	1572 (3.5)	1572 (3.5)	171 (1.6)	171 (1.6)	1401 (4.1)	1401 (4.1)
Diseases of the nervous system and sense organs	2348 (5.2)	2348 (5.2)	259 (2.5)	259 (2.5)	2089 (6.1)	2089 (6.1)
Diseases of the respiratory system	3665 (8.2)	3665 (8.2)	1119 (10.7)	1119 (10.7)	2546 (7.4)	2546 (7.4)
Diseases of the skin and subcutaneous tissue	1327 (3.0)	1327 (3.0)	382 (3.6)	382 (3.6)	945 (2.8)	945 (2.8)
Endocrine, nutritional and metabolic diseases and immunity disorders	652 (1.5)	652 (1.5)	180 (1.7)	180 (1.7)	472 (1.4)	472 (1.4)
External causes of morbidity	19 (<0.1)	19 (<0.1)	4 (<0.1)	4 (<0.1)	15 (<0.1)	15 (<0.1)
Infectious and parasitic diseases	1320 (2.9)	1320 (2.9)	653 (6.2)	653 (6.2)	667 (1.9)	667 (1.9)
Injury and poisoning	1218 (2.7)	1218 (2.7)	86 (0.8)	86 (0.8)	1132 (3.3)	1132 (3.3)
Mental disorders	367 (0.8)	367 (0.8)	80 (0.8)	80 (0.8)	287 (0.8)	287 (0.8)
Neoplasms	3337 (7.4)	3337 (7.4)	1420 (13.5)	1420 (13.5)	1917 (5.6)	1917 (5.6)
Symptoms, signs, and ill-defined conditions	16,834 (37.5)	16,834 (37.5)	2458 (23.4)	2458 (23.4)	14,376 (41.8)	14,376 (41.8)
Patient disposition						
Discharge from ED	32,977 (73.4)	32,977 (36.7)	-	-	32,977 (95.8)	32,977 (95.8)
Transfer to the general ward	9730 (21.7)	9730 (10.8)	9730 (92.8)	9730 (92.8)	-	-
Transfer to the ICU	757 (1.7)	757 (0.8)	757 (7.2)	757 (7.2)	-	-
Transfer to a referral hospital	77 (0.2)	77 (0.1)	-	-	77 (0.2)	77 (0.2)
Departure from ED against medical advice	1172 (2.6)	1172 (1.3)	-	-	1172 (3.4)	1172 (3.4)
Private discharge from ED	48 (0.1)	48 (0.1)	-	-	48 (0.1)	48 (0.1)
Deadly discharge from ED	148 (0.3)	148 (0.2)	-	-	148 (0.4)	148 (0.4)

ED = emergency department; PRIG = preintervention group; POIG = postintervention group; SD = standard deviation; ICU = intensive care unit.

## Data Availability

Not applicable.

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
