# Peer review of "Impact of New Bed Assignment Information System on Emergency Department Length of Stay: An Effect Evaluation for Lean Intervention by Using Interrupted Time Series and Propensity Score Matching Analysis"

_ijerph, 2022, doi:10.3390/ijerph19095364_

Round 1

Reviewer 1 Report

  1. The language of the paper needs a professional editing service. There are many grammatical, spelling and word choice errors. A lot of the sentences are unintelligible. The authors should ask a native English speaker or use professional language editing.
  2. The contribution of this study is weak. We don't know what BAIS is, and the authors are reporting in this study that it has worked to make ER processes more efficient. But what exactly has caused the improvement? What does the present study tell us? How can we use the findings in this study to improve operations in other hospitals?
  3. Using a new IT system will improve the efficiency of operations, in any context and in any kind of operations. Why should the reader be interested in the results of this study? What is interesting and new about it? I don't see that in this paper.
  4. The abstract is cryptic. When I read it, I have no idea what this paper is all about. The abstract should be a very short synopsis of what the paper is about, in plain English; not in bullet or Q/A format.
  5. The paper is highly disorganized. Concepts are just thrown in without  proper explanation. Tables are not well prepared. They are hard to read. For instance, what is ±SD? What is "French leave"? Etc.

Reviewer 2 Report

The manuscript is well written an presents an interesting case of process analysis and improvement with focus on the bed management.

I would suggest the following revision:

- par. 4.3: the comparison with literature should be improved and make more meaningful. A table or a structured analysis of the literature would be beneficial.

- an economic assessment accompanying the bed management evaluations could be beneficial to have a better view of the impact of the proposed work.

Reviewer 3 Report

Thank you for the opportunity to review the paper titled: Impact of New Bed Assignment Information System on Emergency Department Length of Stay: A Before-and-After Study

I have some comments about the work:

1. In material and methods, it is necessary to make a broader description of BAIS. Although, a description is made considering that it is limited.
2. Could the authors, please mention the ethical aspects of the work.
3. In point 3.1 the comparison of the characteristics of the groups is described. However, it is not stated whether these comparisons were significant or not. The authors could indicate it in the text or in Table 1.
4. Please add the values ​​of n in table 2.
5. Taking into account the size of the sample, do the authors consider that using non-parametric statistics is the most appropriate?
6. After BAIS implementation, were all patients assigned to the POIS group? That is, after the implementation of the BAIS, all hospital patients were assigned by this method? How did the authors control for this bias if the answer is yes?

Round 2

Reviewer 1 Report

The authors improved the paper significantly after a revision.